# A Novel Iodine–Dextrin Complex Exhibits No Acute or Subacute Toxicity and Enhances Azithromycin Efficacy in an LPS-Induced Sepsis Model

**DOI:** 10.3390/pharmaceutics17081040

**Published:** 2025-08-11

**Authors:** Nailya Ibragimova, Arailym Aitynova, Seitzhan Turganbay, Marina Lyu, Alexander Ilin, Karina Vassilyeva, Diana Issayeva, Tamari Gapurkhaeva, Arkadiy Krasnoshtanov, Galina Ponomareva, Amir Azembayev

**Affiliations:** JSC Scientific Center for Anti-Infectious Drugs, Almaty 050060, Kazakhstan; nailya73@mail.ru (N.I.); mlyu@mail.ru (M.L.); ilin_ai@mail.ru (A.I.); karinavas12345@gmail.com (K.V.); diana.isaeva.99@inbox.ru (D.I.); versal-85@mail.ru (T.G.); arkada_k@inbox.ru (A.K.); paula1974@mail.ru (G.P.); amirakan@mail.ru (A.A.)

**Keywords:** iodine–dextrin complex, acute toxicity, subacute toxicity, pharmacokinetic study, lipopolysaccharide, sepsis, mice

## Abstract

**Background/Objectives:** Our work was designed to study the physicochemical properties, safety profile, pharmacokinetics, and prophylactic efficacy of an original iodine–dextrin-based pharmaceutical formulation (PA), both alone and in combination with azithromycin (AZ), in a murine model of LPS-induced sepsis. **Methods/Results:** UV–vis and ^1^H-NMR spectroscopy confirmed the formation of a stable iodine–dextrin complex, with triiodide anions stabilized by hydrogen bonding and donor–acceptor interactions. No clinical signs of acute toxicity were observed at doses up to 5000 mg/kg, and subacute administration (62.5 and 125 mg/kg) showed no adverse effects on hematological or biochemical parameters. A mild, non-pathological enlargement of thyrocytes and parallel increases in TSH, T3, and T4 levels were observed at 125 mg/kg, consistent with physiological adaptation to iodine. Pharmacokinetic analysis revealed high oral bioavailability (~92%), prolonged half-life (~21 h), and wide tissue distribution with low clearance. In the sepsis model, pretreatment with AZ+PA alleviated clinical symptoms, maintained body weight, and significantly improved hematological parameters, reducing WBCs and CRP levels. The combination also decreased plasma IL-6 and TNF-α concentrations more effectively than either agent alone, indicating a synergistic anti-inflammatory effect. Histological analysis confirmed that PA, particularly in combination with AZ, mitigated LPS-induced tissue injury in the liver, kidney, and lungs. **Conclusions:** These findings suggest that PA is a safe, bioavailable compound with immunomodulatory properties that enhance azithromycin’s protective effects during systemic inflammation. This supports its potential use as a prophylactic agent in clinical settings, such as preoperative immune modulation to prevent sepsis-related complications.

## 1. Introduction

The high distribution of antimicrobial resistance has become a global concern for antimicrobial treatment of infectious disease. Even if existing antimicrobial drugs are being modified, the rate of resistance continue to increase [1]. A discovery of new groups and classes of antibiotics temporarily attenuates this problem; however, the introduction of new antimicrobial drugs into clinical practice provides an adaptation of pathogens to their active pharmaceutical substances, leading to the rapid stimulation of bacterial antibiotic resistance, as well as its propagation [2]. Thus, an urgent need for the implementation of alternative therapeutic strategies has been created.

A novel iodine complex PA has been developed to address the issue of antibiotic resistance distribution among pathogens. Iodine is a chemical element the antimicrobial activity of which is based on the oxidation of amino acids, nucleotides, and fatty acids of microorganisms, thereby disrupting their cellular processes. However, aqueous solutions of iodine tend to be unstable, and due to the halogenic nature of iodine, they are difficult to study [3]. Thus, iodine coordination compounds, where iodine is fixed with organic macromolecules, have more stability and low toxicity [4].

Dextran is a biocompatible and biodegradable polysaccharide widely used in pharmaceutical and biomedical applications as a matrix for the formation of coordination complexes [5]. Its high-water solubility, abundance of hydroxyl groups, and variability in molecular weight make it a versatile platform for the incorporation of various active substances [6]. Owing to its branched structure, dextran is capable of forming stable complexes with iodine, metals, and biomolecules, stabilizing them through hydrogen bonding, electrostatic interactions, and coordination [7,8,9]. This enables improved solubility, bioavailability, and chemical stability of active agents, as well as sustained release. Furthermore, dextran exhibits no toxicity and does not elicit an immune response, making it especially promising for use in drug delivery systems and topical pharmaceutical formulations [10]. These properties collectively support the selection of dextran as a basis for the formation of stable and biocompatible complexes.

Pharmaceuticals formulations that are structurally based on polymeric complexes of iodine, also known as iodophors, were accurately studied for their bactericidal and fungicidal activities. The well-known and widely used one is povidone (PVP-I) [11]. Due to the presence of polymers in its molecular structure, the topical application of PVP-I provides the gradual release of molecular iodine on the surface of the skin, thereby alleviating the iodine toxicity which can come with the skin irritation [12]. Moreover, bacteria have not developed resistance to the activity of iodine in comparison to existing antibiotics [13]. Thus, the molecular binding of iodine with polymeric complexes allows us to create a novel pharmaceutical formulation which can be safely used for the antimicrobial treatment and will also be more cost-effective than existing antimicrobial drugs. In contrast to PVP-I, PA is an iodine binding to natural polysaccharide dextrin, which offers biocompatibility and biodegradability to its chemical structure. The same principle was followed during the development of a previous pharmaceutical formulation FS-1—an original antiviral drug [14]. When adding water to PA, a blue solution is formed due to the intramolecular binding of iodine to dextrin, thereby inducing a chemical reaction of complexation in the presence of potassium iodide. The possibility of the formation of amylose type V during dextrin hydrolysis is not excluded. This mechanism gives PA dark blue coloration and provides the complexation reaction: hydrophobic iodine molecules inside the helix are present together with iodide anions [15]. Due to the fact that the molecular iodine molecule (I_2_) is hydrophobic and, therefore, it is predominantly located inside the hydrophobic helix of amylose. At the same time, other iodine ions do not participate in the incorporation into the iodine–amylose helix but help to dissolve iodine in water and in the creation of combinations of I_2_ and I^−^, including I^n−^, inside the amylose helix [16,17]. In addition, it is molecular iodine that, along with antibacterial properties, also exhibits an immunomodulatory effect of PA [18]. PVP-I does not exhibit such effect and cannot be taken orally in contrast to PA.

The study of pharmacokinetic parameters has been used to predict the efficacy and potential toxicity of new pharmaceutical formulations. As a part of preclinical studies, the pharmacokinetic characteristic, tissue distribution, and excretion will guide the rational use of drugs in clinical trials [19,20]. Therefore, pharmacokinetic studies of PA will help to reveal the pharmacological activity and, hence, promote its clinical application.

A dysregulated host response to bacterial infection leads to organ dysfunction, known as septic shock. It is associated with a high risk of mortality in human population worldwide [21]. Lipopolysaccharide (LPS) is the main part of bacterial membrane and a key agent promoting septic shock via activation of macrophages and secretion of pro-inflammatory cytokines (IL-1β, IL-6 and TNF-α). It is widely used for animal model of sepsis to study the protective and/or therapeutic effects of pharmaceutical formulations [22].

The efficacy of antibiotics against resistant microorganisms can be restored or even enhanced by the action of antibiotic potentiators [23]. This is a promising approach to alternative therapy in infectious disease. Thus, the presented work started with the study of acute and subacute toxicity of PA in order to evaluate its safety and to select a dose for follow-up pharmacokinetic study. Right after the calculation of pharmacokinetic parameters, the antibiotic potentiation efficacy of PA at the dose of 62.5 mg/kg was studied in mice with LPS-induced sepsis.

## 2. Materials and Methods

### 2.1. Chemical Studies

Potassium iodide (Sigma-Aldrich, St. Louis, MO, USA); iodine (Labpharm, JSC “Troitsky iodine plant”, Troitsk, Russia); albumin (BioPharma Plasma, Kyiv, Ukraine); corn-derived dextrin (molecular weight~4000 kDa and deacetylation degree of 95%) from Sigma-Aldrich (St. Louis, MO, USA, ≥99%). The reagents used for the present work were of analytical grade, obtained from commercial sources, and used without further purification.

#### 2.1.1. Synthesis and Structural Formula

An original iodine–dextrin-based complex (PA) was synthesized by the Laboratory of New Substances and Materials of JSC “Scientific Center for Anti-Infectious Drugs” (Almaty, Kazakhstan). The chemical synthesis process involved a coordination reaction resulting in the formation of an iodine–polymer complex. The complex is obtained through two successive reactions in aqueous solution. The first reaction occurs between dextrin and albumin, leading to the formation of an intermediate—a protein–carbohydrate adduct. Then this adduct reacts with polyiodide, leading to the formation of the iodine–dextrin-based complex (PA).

The proposed structural formula of PA is shown in Figure 1.

#### 2.1.2. UV–Vis Spectroscopy

UV–vis spectra of the samples were collected using a LAMBDA-35 UV–Vis spectro-photometer (PerkinElmer, Waltham, MA, USA). The samples were dissolved in a water solution (1 mg/mL), and the solvent was used as a reference. The scanning range was 190–1100 nm [24].

#### 2.1.3. Determination of the Iodine Content

The concentration of halogens (iodine) in the complexes was measured by sodium thiosulfate titration [25]. The concentration of molecular iodine per 1000 g of the complex was measured by Equation (1):(1)$$C_{I_{2}}=\frac{V_{1}\cdot K_{1}\cdot 12.69}{m},$$
where

V_1_—volume of 0.05 M sodium thiosulfate spent for complete titration (mL);

K_1_—correction on sodium thiosulfate concentration in the buffer, for 0.05 M solution K_1_ = 0.5;

m—weight of the complex (g).

The concentration of KI was measured by Equation (2) using titration with silver nitrate:(2)$$C_{KI}=\frac{{(V}_{2}\cdot K_{2}\cdot {(V}_{1}\cdot K_{1})\cdot 16.59}{m},$$
where

V_1_—volume of 0.05 M sodium thiosulfate spent for complete titration (mL);

V_2_—volume of 0.05 M AgNO3 spent for complete titration (mL);

K_1_ = K_2_ = 0.5;

m—weight of the complex (g).

#### 2.1.4. ^1^H NMR-Spectroscopy

^1^H-NMR spectra of the samples were collected using a superconducting Fourier NMR spectrometer JNM-ECA 500 (JEOL, Tokyo, Japan) operating at 500 MHz, and the solvent used was DCl/D2O (1/100, *v*/*v*).

### 2.2. In Vivo Studies and Bioethics

All in vivo studies were carries out in accordance with the “Guide for the Care and Use of Laboratory Animals” and ARRIVE guidelines. The experimental protocol was reviewed and approved by the Ethical Committee of JSC “Scientific Center for Anti-Infectious Drugs” No. 25/1. Animals were maintained at the Animal Facility under conventional laboratory conditions (environment temperature 25 ± 1 °C and the relative humidity 55 ± 5% with light/dark cycle of 12/12 h). They were housed in propylene cages, five animals per cage with free access to standard rodent diet and water ad libitum. Animals were acclimatized for one week before the start of experiments.

#### 2.2.1. Acute Toxicity Study

The acute toxicity study was conducted in accordance with the Test No. 425 of Organization for Economic Cooperation and Development (OECD) Guideline [26]. Ten healthy Swiss albino mice (five male and five female), 6 months old, body weight 28–30 g were used for this experiment. Since the test item was presumably low-toxicity, the higher doses of 2000 mg/kg and 5000 mg/kg were selected for the acute toxicity test. On Day 0, the animals received a single dose of PA per os, then they were observed for fourteen days for any signs of toxicity. A solution of PA with water was prepared shortly prior to administration. On Day 15, animals were euthanized and subjected to the gross examination of internal organs.

#### 2.2.2. Subacute Toxicity Study

The subacute toxicity study was conducted in accordance with the Test No. 407 of Organization for Economic Cooperation and Development (OECD) Guideline [27]. Forty healthy Swiss mice (twenty male and twenty female), 6–8 weeks old, body weight 24–26 g were used for this experiment. They were divided into four groups of ten animals per group (five male and five female): 1st group—control, 2nd group receiving 62.5 mg/kg PA, 3rd group receiving 125 mg/kg PA, 4th group receiving 250 mg/kg PA. A solution of PA with water was prepared daily shortly prior to repeated administration. Animals were observed daily and weighed weekly for 28 days in order to detect mortality or subacute toxicity signs.

#### 2.2.3. Pharmacokinetic Study

Healthy Swiss albino mice of both sexes, 10–12 weeks old, body weight 22–26 g were used for the pharmacokinetic study. They were orally administered PA at the dose of 62.5 mg/kg, which was selected in reference to the subacute toxicity study. Serial blood collection was conducted at 10, 15, 30, 45 min; 1, 2, 4, 6, 8, 24, 30, 48 h after administration of PA. The Inductively Coupled Plasma Mass Spectrometry (ICP-MS) was used to determine the quantity of iodine at the selected time-points. The following pharmacokinetic parameters were determined: total area under the serum concentration–time curve (AUC_total_), the highest concentration of a drug in the blood (C_max_), time for a drug to reach the maximum concentration after its administration (t_max_), long elimination half-life (t_1_/_2_), systemic clearance (Cl_s_), volume of distribution (V_β_), elimination rate constant (k_e_).

#### 2.2.4. LPS-Induced Abdominal Sepsis Model

Twenty healthy Swiss albino mice, 10–12 weeks old (both sexes, body weight 22–26 g) were used in this experiment. They were randomly divided into four groups: 1st—LPS-group received no therapy, 2nd—LPS-group received azithromycin, 3rd—LPS group received PA, 4th—LPS group received azithromycin and PA. The sepsis model was induced by an intraperitoneal injection of LPS (1 mg/kg) to mice 1 h before the appropriate therapy [28]. Azithromycin was given to mice at the dose of 25 mg/kg, which was based on its concentration in water (0.625 mg/mL) and the average water consumption and bodyweight of animals [29,30]. The dose of PA was selected to be 62.5 mg/kg in reference to subacute toxicity and pharmacokinetic studies. For the evaluation of prophylactic effect, PA and azithromycin were administered 1 h before the induction of LPS-sepsis. The modified Murine Sepsis Score (MSS) was used to assess the severity of sepsis in mice [31]. Animals were observed and scored (0–3) for the following markers: appearance, level of consciousness, activity, response to stimulus, eyes, and respiratory quality for 24 h. The MSS scoring was performed independently by two blinded investigators for each sepsis component, and recorded at 1, 2, 4, 6, 8, 10, 12, 14, 16, 18, 20, 22, 24 h. Then, the scores of each mouse were used to calculate the mean score for the group. Animals were also weighed before and on Days 3, 7, and 10 after the induction of sepsis. At the end of the experiment, animals were euthanized and peritoneal lavage was obtained for immunological analysis of the levels of IL-6 and TNF-α.

### 2.3. Euthanasia and Sample Collection

At the end of each experiment animals were anesthetized by deep isoflurane narcosis. Blood was collected via retroocular sinus puncture in clot-activator tubes, then mice were sacrificed via cervical dislocation. Blood samples were centrifuged at 3000 rpm for 10 min, then plasma was separated for biochemistry analysis. Analysis of hematological parameters was conducted on Z52 VET automatic hematology analyzer (Zytopia Ltd., Chongqing, Dadukou, China). Plasma biochemistry was measured using a fully automated benchtop chemistry analyzer A25 (BioSystems, Troisdorf, Koln, Germany) with special kits according to the manufacturer’s instructions. Internal organs were excised and fixed in buffered 10% formaldehyde for 24 h. The serum triiodothyronine (T3), thyroxine (T4), and thyroid-stimulating hormone (TSH) were investigated through enzyme-linked immunosorbent assay (ELISA), according to the manufacturer protocols. The quantity of iodine in blood was analyzed by Inductively Coupled Plasma Mass Spectrometry (ICP-MS) system. The immunological study was conducted through ELISA according to the protocols provided by kits. Then tissue sections were dehydrated in alcohol, embedded in paraffin, sectioned into 3.0 μm fragments and stained with hematoxylin-eosin. Histological evaluation was performed under a ZEISS Axio Scope A1 light microscope (Carl Zeiss, Oberkochen, Germany).

### 2.4. Statistical Analysis

Data are presented as mean ± standard deviation (SD). A two-way analysis of variance (ANOVA) was performed for comparisons between groups, followed by the Bonferroni post hoc test to assess statistical significance. Statistical analyses were conducted using GraphPad Prism software, version 6.0 (GraphPad Software Inc., San Diego, CA, USA). A *p*-value of less than 0.05 was considered statistically significant.

## 3. Results

### 3.1. Physicochemical Analysis of PA

#### 3.1.1. Properties and Stability

The physicochemical properties of PA are summarized in Table 1. It appeared as a dark gray to dark brown powder, likely due to polyiodide formation with dextrin. The synthesis resulted in a high percentage yield and a sharp melting point, indicating structural stability. Notably, the complex showed high solubility in water, attributed to the hydrophilic nature of dextrin and stabilization of iodine in the polymer matrix. These properties make the complex suitable for further pharmaceutical applications.

The stability of PA largely depends on the chemical composition and storage conditions [32]. Studies of the stability of the PA were carried out on three laboratory series in accordance with the requirements of the regulatory documents of the European pharmacopeia and ICH, according to the following quality indicators: description, solubility, melting, temperature, pH, and quantitative determination of iodine and iodide ions. Over the studied storage period (0, 3, 6, 9, 12 months), PA showed a constant composition and no significant changes in the quality indicators of the coordination compound (Table 2).

#### 3.1.2. UV–Vis Spectral Analysis

UV–visible spectroscopic analysis confirmed the formation of a coordination complex in the PA. The UV spectrum of dextrin (Appendix A) exhibited a weak absorption band at ~278 nm, attributed to n→π transitions involving heteroatoms such as oxygen in the polysaccharide structure, in agreement with data reported by Zhang et al. [33]. Albumin displayed a strong absorption maximum at 204.27 nm, corresponding to π→π transitions in peptide bonds (–CO–NH–), and additional peaks at 255.68 and 278.11 nm, related to n→π transitions of aromatic amino acids (tryptophan and tyrosine) (Appendix A). These features confirm the presence of amino acid residues in the structure, though albumin in the complex likely exists in a fragmented form involved in coordination, rather than in its native state. The I_2_ + KI solution showed characteristic absorption bands at 195.81 and 223.68 nm (molecular iodine and its interaction with iodide ions), as well as at 287.05, 350.91, and 452.29 nm, indicative of triiodide ions (I_3_^−^)—the predominant species in the presence of excess KI (Appendix A). However, in the absence of a stabilizing matrix, such species are generally unstable and prone to dissociation.

The spectrum of the PA (Appendix A) substance revealed enhanced bands at 196.51 and 224.85 nm, along with additional absorptions at 288.60, 349.10, 465.86, and 561.04 nm. The latter bands were absent in the spectra of the individual components and are typical for iodine-based coordination complexes involving donor–acceptor interactions and charge-transfer effects (LMCT). Notably, the bands in the 465–561 nm region are consistent with stabilized polyiodide species in coordination environments, as reported by Lee et al. [34].

These spectral changes confirm the formation of a stable iodine–dextrin-based semi-organic complex, where triiodide anions are stabilized through interactions with the hydroxyl groups of dextrin and functional moieties of albumin (amino, carboxyl, and aromatic residues). The resulting polymer system exhibits enhanced electronic transitions and long-wavelength absorption, characteristic of structured ligand–iodine assemblies with both hydrogen bonding and halogen-type interactions.

#### 3.1.3. ^1^H-NMR Spectroscopic Characterization

As shown in Figure 2, the ^1^H-NMR spectrum of the PA substance reveals characteristic signals confirming the presence of dextrin and albumin-derived fragments involved in complex formation. The resonance at 5.224 ppm corresponds to the anomeric proton (H-1) of dextrin, indicating preserved α-1,4-glycosidic linkages. Multiplets in the 3.1–3.8 ppm range reflect protons at C2–C6 of glucose units and overlapping signals from amino acid side chains.

A signal at 4.484 ppm is attributed to α-protons and exchangeable NH groups of peptide fragments. The downfield shift at 5.063 ppm likely results from coordination between molecular iodine (I_2_) and glycosidic oxygen atoms. Additional peaks at 3.245 and 3.111 ppm suggest electronic perturbations in CH_2_ groups of dextrin due to iodide or polyiodide interactions. Signal broadening and chemical shift variations, compared to the individual components [35], indicate the formation of a stable iodine–polymer coordination complex via hydrogen bonding and donor–acceptor interactions. These spectral features confirm the supramolecular structure and coordinated nature of the PA substance.

### 3.2. Toxicity Studies

#### 3.2.1. Acute Toxicity Test

A single oral administration of PA at the doses of 2000 mg/kg and 5000 mg/kg did not induce any behavioral changes and clinical signs in mice of both sexes. No alteration in internal organs was observed after euthanasia. The tendency of the bodyweight change in mice of both sexes is presented in Table 3.

There was no significant difference between mice of the control group, and the group received PA at the doses of 2000 mg/kg and 5000 mg/kg, all animals demonstrated normal rate of weight gain (Table 3).

The effect of acute administration of PA at the doses of 2000 mg/kg and 5000 mg/kg on hematological parameters of mice is presented in Figure 3.

No significant difference in levels of red and white blood cells, lymphocytes, monocytes, granulocytes, hemoglobin, platelets, and the percentage of hematocrit between the control group and the group received PA at the doses of 2000 and 5000 mg/kg was observed (Figure 3).

Analysis of plasma biochemistry showed that the acute administration of PA did not alter hepatic (ALT, AST) and renal function (urea, creatinine) as shown in Figure 4.

No significant difference in levels of liver transaminases, urea, and creatinine between the control group and the group received PA was observed, all parameters remained within the reference range (Figure 4).

Overall, no clinical symptoms of acute toxicity in mice were noted during the 14-day observation period. Our findings suggest that oral administration PA does not cause acute toxicity. Also, since no mortality among animals was noted, the LD_50_ was impossible to calculate.

#### 3.2.2. Subacute Toxicity Test

PA was repeatedly administered to mice at doses of 62.5 and 125 mg/kg for 28 days. Animals displayed no death and no clinical signs of subacute toxicity until the end of the experiment. The weight gain and food and water consumption were similar to the control group. Results of body weight measurements are presented in Table 4.

No significant difference was noted between groups receiving PA at different doses and the control group. Animals displayed a similar rate of weight gain (Table 4).

Hematological parameters of animals treated with PA are similar to the control group. No significant difference was noted between levels of red and white blood cells, as well as hemoglobin values and the percentage of hematocrit of treated and control groups of mice (Figure 5).

Animals receiving repeated doses of PA displayed no significant alteration in plasma biochemistry markers (Figure 6). The subacute administration of PA did not affect hepatic (ALT, AST) and renal function (urea, creatinine).

Histological examination of internal organs: heart, liver, lungs, kidney, spleen and thyroid gland of mice administered PA at doses of 62.5 and 125 mg/kg in comparison to the control group was conducted. Microscopic photographs are presented in Figure 7.

Normal cardiac muscle fibers arranged in bundles with clear striations, and no evidence of necrosis, inflammation, or edema were observed during histological examination of control heart tissue (Figure 7A). Similarly to the control, PA 62.5 and 125 mg/kg groups demonstrated normal architecture and intact muscle fibers (Figure 7B,C), indicating that PA does not induce cardiotoxic effects at studied doses, which is important given the sensitivity of cardiac tissue to systemic agents [36]. Liver sections maintained intact lobular architecture with no signs of hepatocellular degeneration, but with mild degree of Kupffer cell activation (Figure 7D,E) and a slight prominence of nucleoli at 125 mg/kg (Figure 7F), likely reflecting increased metabolic activity rather than toxicity [37]. Lung tissue showed open alveoli and thin septa with no signs of exudate, hemorrhage, or inflammatory infiltrates (Figure 7G–I), consistent with an absence of pulmonary irritation or oxidative injury commonly observed with some iodine-based agents [38]. In kidneys, glomeruli and tubules appeared normal, with no vacuolization or epithelial sloughing (Figure 7J–L), indicating that the gradual release of iodine from the dextrin matrix did not overload renal clearance mechanisms [39]. Spleen sections retained balanced red and white pulp zones with intact lymphoid follicles (Figure 7M–O), suggesting no immunosuppressive or hyperstimulatory effects [40]. In the thyroid gland, follicles were filled with colloid and lined with intact epithelium in the control (Figure 7P) and 62.5 mg/kg PA group (Figure 7Q); however, at 125 mg/kg, mild enlargement of thyrocytes was noted, with the epithelium transitioning from cuboidal to slightly columnar (Figure 7R). This change correlated with the observed parallel increases in serum TSH, T3 and T4 levels, that are presented in Figure 8. This may be due to the physiological adaptation to increased iodine supply rather than pathological hyperplasia [41,42]. Importantly, colloid content was preserved, and no signs of follicular atrophy or lymphocytic infiltration were present.

As shown in Figure 8, an increase in levels of T3, T4, and TSH was observed in mice that received the dose of 125 mg/kg (*p* < 0.001) which is in agreement with the histology of the thyroid gland. An enlargement of thyrocytes may be due to TSH growth effect on follicular cells, which stimulates thyroid follicular cells to release thyroid hormones in the form of T3 or T4. However, a parallel increase in all hormones was observed indicating that feedback between TSH and T3/T4 in the pituitary–thyroid axis remained intact [43]. This suggests that the thyroid gland remained functionally responsive and indicates a compensated endocrine state rather than a pathological disruption. This phenomenon has been observed in other non-toxicological contexts, such as cold adaptation or increased energy turnover, where elevated thyroid hormones occur in the absence of glandular pathology [44]. Moreover, severe hyperthyroidism is often associated with elevated levels of ALT and AST, which if long-term may cause significant liver damage [45]. However, no pathological deviation was observed in levels of liver transaminases (Figure 6), indicating a normal metabolism of amino acids and glucose, thereby normal function of the liver. Moreover, the histological structure of hepatic tissue appeared to be intact (Figure 7D–F), which is also important since the liver plays a crucial role in conversion of T4 to T3, as well as conjugation and excretion of thyroid hormones [46]. The same occurred with kidneys, that demonstrated a normal histological structure (Figure 7J–L) along with levels of urea and creatinine within the normal range for mice (Figure 6) [47]. Severe hyperthyroidism is also accompanied by elevated blood urea nitrogen and decreased serum creatinine indicating an impaired renal function [48]. Thus, our results demonstrate that moderate dosing of PA is well tolerated, even in metabolically active organs such as the liver and kidney. In the heart, muscle fibers remained well aligned and intact, suggesting no cardiotoxic effects (Figure 7A–C). Similarly, lung tissue retained its normal histoarchitecture across all groups. Alveolar spaces were open, septa remained thin and free of inflammatory cell infiltration or fibrosis, and no edema or congestion was observed (Figure 7G–I). This indicates that PA does not elicit pulmonary inflammation or oxidative injury, which are common concerns for systemically active agents with high tissue penetration [49]. The spleen, a highly vascular immune organ, showed no evidence of lymphoid depletion, white pulp atrophy, or congestion (Figure 7M–O). This is significant given that some antimicrobial drugs can lead to immune cell turnover or lymphoid stress [50]. A normal histological structure of spleen indicates that PA does not cause immunotoxicity or disrupt splenic immune homeostasis. Overall, the preserved architecture across all organs—alongside the mild, adaptive thyroid response—indicates that PA is histologically well-tolerated and does not disrupt structural integrity, reinforcing its potential as a safe, slow-releasing iodine–dextrin based formulation.

### 3.3. ICP-MS and Pharmacokinetic Parameters

Blood collected at 10, 15, 30, 45 min and 1, 2, 4, 6, 8, 24, 30, 48 h was analyzed through ICP-MS system for determination of the quantity of iodine, results are presented in Figure 9.

As shown in Figure 9, the peak concentration of iodine in blood of mice was observed two hours after the oral administration of 62.5 mg/kg PA and calculated to be 1262.89 µg/L. A gradual decrease was noted for the upcoming time-points, reaching the minimal threshold of 107.27 µg/L 48 h after administration of 62.5 mg/kg PA.

According to Table 5, the pharmacokinetic analysis following oral administration of 62.5 mg/kg PA revealed a C_max_ of 2437.8 μg/mL at t_max_ of 4 h, and a long elimination half-life (t_1/2_ = 21.1 h). The high oral bioavailability (F_abs_ 91.96%) and low systemic clearance (0.26 L/kg) suggest efficient absorption and sustained systemic exposure. These results were derived using ICP-MS quantification of iodine, taking advantage of iodine’s excellent detectability, low background levels in biological matrices, and high mass-to-charge ratio, which makes it ideal for tracing iodine-containing compounds in vivo [51,52]. The volume of distribution (V_β_ = 7.8 L/kg) indicates extensive tissue penetration, which aligns with previous studies showing that iodine-containing molecules often accumulate in soft tissues due to their partial lipophilicity and interaction with carrier proteins [53]. Using iodine as a tracer element for pharmacokinetic assessment has been validated in multiple preclinical studies of iodinated drugs and radiocontrast agents. It offers a sensitive, non-radioactive approach for tracking compound fate in vivo [54]. Thus, the current results reflect both the effective systemic delivery of PA and the reliability of iodine-based ICP-MS analysis in pharmacokinetic profiling.

### 3.4. Effect on LPS-Induced Sepsis

All animals survived up to the end of the experiment, however, mice that did not receive either separate or combined therapy displayed more vigorous features of the development of sepsis. Figure 10 represents a MSS six-parameter assessment of sepsis severity in mice following LPS injection and prophylactic treatment with AZ, PA, and their combination (AZ+PA). Parameters that were assessed: appearance, level of consciousness, activity, response to stimulus, eye appearance, and respiration quality over a 24 h period.

As expected, NC animals remained stable across all parameters. In contrast, the PC group elevated scores for all metrics, reaching near-minimal values by 12–16 h, indicating the full development of systemic sepsis symptoms. Mice treated with AZ or PA alone showed moderate improvement in clinical signs compared to PC, with slower decline rates and higher average scores in all MSS parameters, particularly for level of consciousness, activity, and respiration quality. The improvement with PA is notable in maintaining higher stimulus response and eye appearance scores over time, indicating preserved neurological and systemic stability. The AZ+PA combination group showed the most consistent and significant protection across all parameters. Scores remained significantly lower than PC and lower than either AZ or PA alone, particularly in the appearance, consciousness, and activity domains, where the curves are visibly and consistently separated from those of other groups over the 24 h period.

In addition to the MSS method of the severity of sepsis assessment, animals were weighed prior to the injection of LPS (Day 0) and on Days 3, 7, and 10 (Table 6). As a result, mice of the positive control group showed rapid weight loss on Days 3, 7, and 10 (*p* < 0.05). This is likely due to suppressed eating and drinking as a clinical symptom of murine septic shock [55]. Animals receiving AZ and PA separately showed slower rate of weight loss on Days 3 and 7, but a little gain on Day 10. However, the AZ+PA group demonstrated normal food and water intake, thereby steady weight gain on those days (*p* < 0.05). This may be due to alleviation of the acute phase of the development of sepsis.

Analysis of the hematological parameters was conducted in order to study the RBCs count and parameters such as hemoglobin, hematocrit, sedimentation rate; as well as levels of WBCs, lymphocytes and C-reactive protein (Figure 11).

LPS-induced sepsis is well known to disrupt hematopoiesis and trigger systemic inflammation through the activation of innate immune pathways, leading to alterations in RBCs indices, WBCs profiles, and acute-phase reactants such as C-reactive protein (CRP) [56]. As shown in Figure 11, the positive control group receiving LPS without treatment showed a significant decrease in RBC count, hemoglobin, and hematocrit (*p* < 0.01), consistent with inflammation-driven suppression of erythropoiesis and iron sequestration [57]. In contrast, combined treatment with AZ+PA significantly restored these values (*p* < 0.05 to *p* < 0.01), indicating protection of red blood cell homeostasis. Similarly, the elevated ESR, WBCs, monocytes, granulocytes, and lymphocytes in the PC group (*p* < 0.01 to *p* < 0.001) reflected a typical acute inflammatory response [58]. AZ+PA significantly reduced these markers (*p* < 0.05 to *p* < 0.01), suggesting a systemic anti-inflammatory effect. The combination group (AZ+PA) showed the most consistent normalization of leukocyte populations, supporting a synergistic immunomodulatory effect. Importantly, CRP levels, a hallmark marker of acute-phase inflammation, were significantly elevated in the PC group (*p* < 0.01). While neither AZ nor PA alone significantly reduced CRP, their combination (AZ+PA) did achieve a notable decrease (*p* < 0.05), suggesting enhanced suppression of pro-inflammatory cytokine activity when used together [59,60]. Elevated platelet counts in the PC group (*p* < 0.01) are a hallmark of early sepsis-related hypercoagulability. Treatment groups exhibited a dose-dependent normalization (*p* < 0.01 to *p* < 0.001), with AZ+PA showing the most substantial reduction, possibly reflecting restoration of vascular integrity and dampening of thrombo-inflammatory pathways [61].

ELISA was conducted on peritoneal wash obtained from mice after euthanasia. Values of the pro-inflammatory cytokines IL-6 and TNF-α that are primarily specific to the septic process are presented in Figure 12.

LPS-induced sepsis triggers a systemic cytokine storm marked by sharp increases in pro-inflammatory mediators, particularly interleukin-6 (IL-6) and tumor necrosis factor-alpha (TNF-α), both of which play central roles in sepsis pathogenesis and organ dysfunction [62,63]. As shown in Figure 12, the positive control group demonstrated significant elevations in IL-6 and TNF-α levels (*p* < 0.001), reflecting a robust inflammatory response. Treatment with AZ or PA alone significantly reduced both cytokines (*p* < 0.05), and the combination therapy (AZ+PA) further lowered IL-6 (*p* < 0.01) and TNF-α levels, suggesting a synergistic anti-cytokine effect. AZ is known to suppress pro-inflammatory cytokine production by modulating macrophage function and inhibiting NF-κB activation [64], while molecular iodine has demonstrated antioxidant and cytokine-modulating activity, possibly through interaction with cellular redox systems and PPAR-γ signaling [65,66]. The fact that AZ+PA treatment produced the greatest reduction in both cytokines indicates a complementary mechanism of action, where PA enhances the immune-regulating potential of azithromycin. These results align with findings from experimental and clinical studies in which macrolides and iodine-based agents decreased circulating IL-6 and TNF-α levels in inflammatory states [67,68].

Histological evaluation of liver, kidney, and lung tissues revealed that both PA and AZ, individually and in combination, mitigated tissue damage induced by LPS-triggered systemic inflammation, microscopic photographs are presented in Figure 13.

In the liver, the positive control group that received LPS but no treatment—exhibited classic sepsis-associated histopathology, including hepatocyte disorganization, cytoplasmic vacuolization, and inflammatory infiltrates (Figure 13A). Azithromycin alone moderately improved hepatic architecture, reducing inflammatory cell presence (Figure 13B), while the PA group showed more preserved lobular organization and reduced cellular damage (Figure 13C). Most notably, combined treatment with AZ and PA showed more restored hepatic morphology, with near-normal hepatocyte arrangement and minimal inflammatory infiltration (Figure 13D), suggesting a synergistic hepatoprotective effect, potentially due to an antimicrobial, antioxidant and immunomodulatory effects of PA complementing antibiotic properties of AZ [69,70]. The same was observed in kidney histology. The positive control group displayed tubular epithelial damage, glomerular shrinkage, and interstitial inflammation, typical of septic nephropathy (Figure 13E). AZ treatment partially reduced tubular injury and glomerular damage (Figure 13F). In contrast, PA alone preserved overall nephron architecture with reduced inflammation (Figure 13G). The AZ+PA combination offered the most protection, with structurally intact glomeruli and tubules and minimal interstitial changes (Figure 13H), supporting the hypothesis that PA may mitigate LPS-induced oxidative renal damage, consistent with previous reports on iodine’s tissue-stabilizing effects [71]. In the lungs, the LPS control group showed alveolar wall thickening, inflammatory infiltrates, and disrupted airspaces, all features of acute lung injury (Figure 13I). AZ reduced inflammatory cell presence, though alveolar thickening persisted (Figure 13J). PA treatment maintained more open alveolar spaces and reduced septal congestion (Figure 13K), while the AZ+PA group demonstrated near-complete restoration of pulmonary architecture (Figure 13L), suggesting a combined anti-inflammatory and barrier-stabilizing effect. Given the known anti-inflammatory properties of azithromycin in sepsis and iodine’s reported role in maintaining epithelial integrity, attenuating oxidative stress, and modulating immunity [72,73], the combined treatment effect is biologically plausible.

In addition to the histological results, the progressive bodyweight loss, plummeted levels of RBCs and hemoglobin along with elevated levels of leukocytes as well as of the pro-inflammatory cytokines IL-6 and TNF-α in mice that received no treatment can justify the effectiveness of the prophylactic combined treatment of AZ+PA, that lead to the alleviation of those symptoms. The low food and water intake accompanied with weight loss is one of the determinants for murine sepsis, along with stationary behavior and physical activity [74]. However, no signs of bodyweight loss, rather maintenance on Day 3 and a little bodyweight gain on Days 7 and 10 were observed in the group of mice that received AZ+PA (*p* ≤ 0.05), as well as any deviations in social behavior. Changes in RBCs parameters, particularly hemoglobin and hematocrit, can be important indicators of sepsis severity and prognosis. LPS can alter or even suppress erythropoiesis in the bone marrow and induce the rupture of RBCs, leading to a decrease in hemoglobin and RBCs count [75]. All of this can potentially lead to impaired blood flow and tissue ischemia, as well as induce the development of anemia which will be especially critical in patients with sepsis [76]. Septic shock triggers a cascade of events involving leukocyte activation, adhesion to blood vessel walls and their migration into tissues [77]. These processes, while crucial for fighting infection, can become dysregulated when excessive activation of leukocytes, including neutrophils and macrophages, cause them to uncontrollably release inflammatory mediators like IL-6 and TNF-α which are the most prominent cytokines released in response to LPS [78,79]. PA and azithromycin were administered to mice 1 h prior the intraperitoneal injection of LPS in order to evaluate the prophylactic effect of PA in combination with antibiotics. The efficacy of the combination of AZ+PA was shown to stimulate the host immunomodulatory response, which was observed in all mice parameters described above just before the onset of sepsis. Thus, results of this experiment can justify the prophylactic approach of treatment with PA and azithromycin, which in clinical settings may be used prior to surgical manipulations.

## 4. Conclusions

This study aimed to characterize the chemical structure and physicochemical properties, evaluate the safety profile and pharmacokinetic parameters, and assess the prophylactic potential of a novel pharmaceutical formulation PA, an original iodine–dextrin-based complex. No mortality or clinical signs of toxicity were observed in mice treated with PA at single oral doses of 2000 mg/kg and 5000 mg/kg. This indicates that PA is non-toxic at high doses, supporting its safety for further pharmacological evaluation. Repeated oral administration of PA for 28 days at doses of 62.5 mg/kg and 125 mg/kg caused no histological damage in vital organs. A mild, parallel increase in TSH, T3, and T4 levels at the highest dose suggests an adaptive thyroid response, with the pituitary–thyroid axis remaining intact. PA demonstrated high oral bioavailability (~92%), long plasma half-life (~21 h) and broad tissue distribution with low clearance. These parameters support the sustained systemic availability of iodine from the dextrin complex without accumulation-related toxicity. PA in combination with azithromycin significantly enhanced the efficacy of antibiotic azithromycin, which was demonstrated in the levels of inflammatory cytokines, as well as attenuated LPS-induced liver, kidney, and lung damage. These effects reflect anti-inflammatory and immunomodulatory properties, enhancing the therapeutic potential of standard antibiotics in systemic inflammation. Overall, findings of the presented work demonstrate that the iodine–dextrin complex PA not only lacks toxicity but also exhibits an enhancement effect on azithromycin during sepsis, likely via its antimicrobial and immunomodulatory mechanisms.

## Figures and Tables

**Figure 1 pharmaceutics-17-01040-f001:**
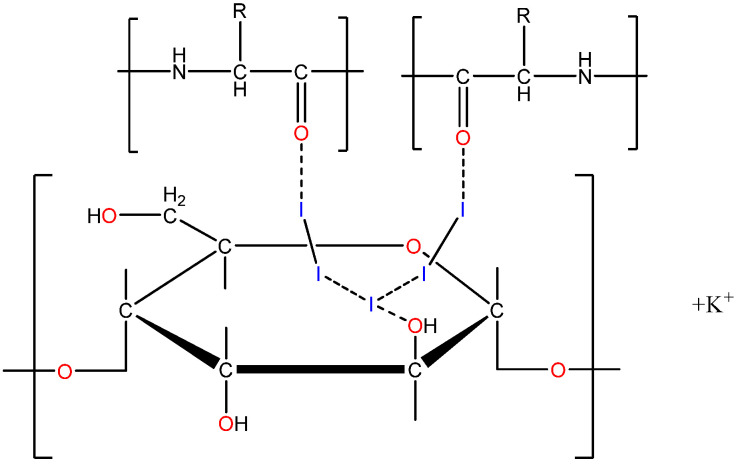
Structural formula of iodine–dextrin-based complex (PA).

**Figure 2 pharmaceutics-17-01040-f002:**
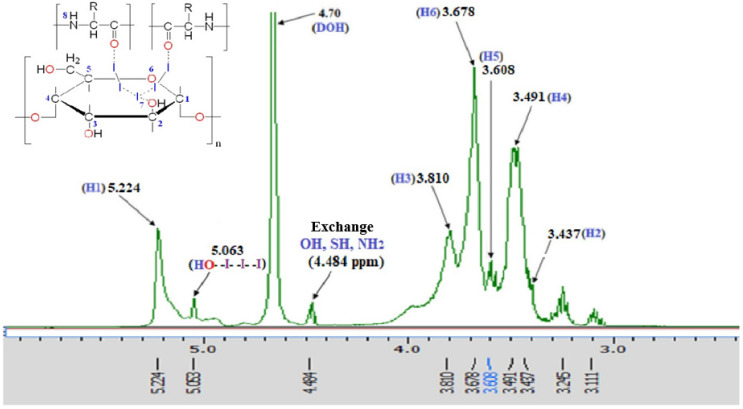
^1^HNMR spectrum of PA.

**Figure 3 pharmaceutics-17-01040-f003:**
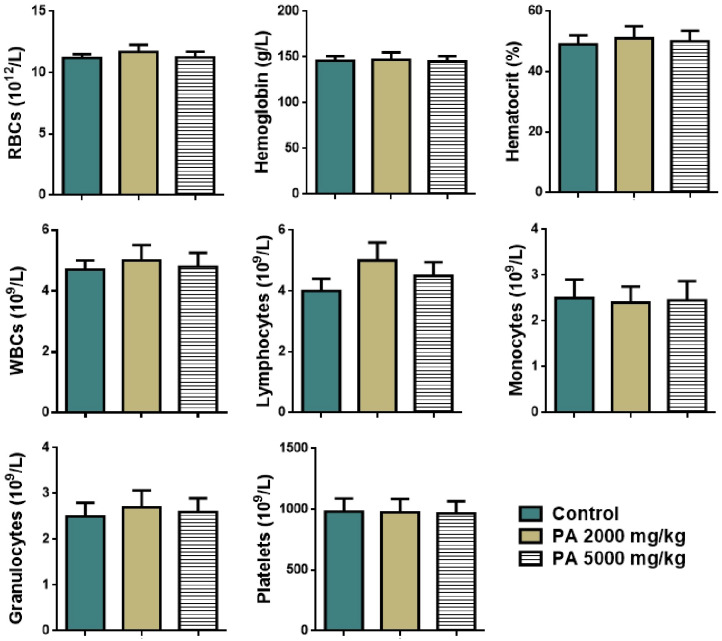
Hematological parameters of mice during the acute toxicity study. Note: RBCs—red blood cells, WBCs—white blood cells.

**Figure 4 pharmaceutics-17-01040-f004:**
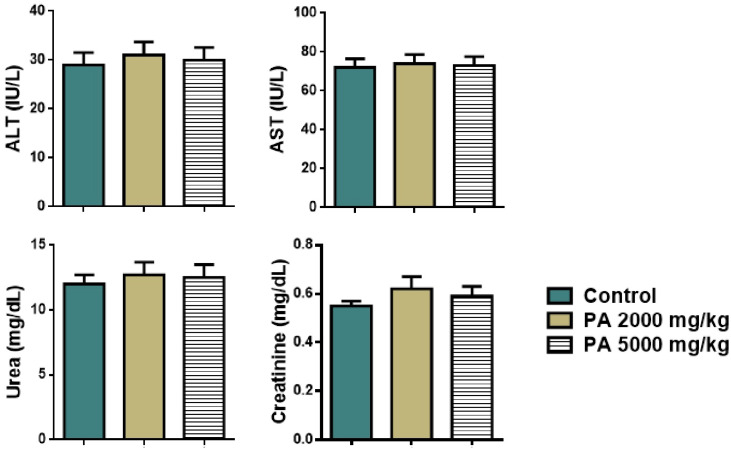
Plasma biochemistry of mice during the acute toxicity study. Note: ALT—alaninaminotransferase, AST—aspartataminotransferase.

**Figure 5 pharmaceutics-17-01040-f005:**
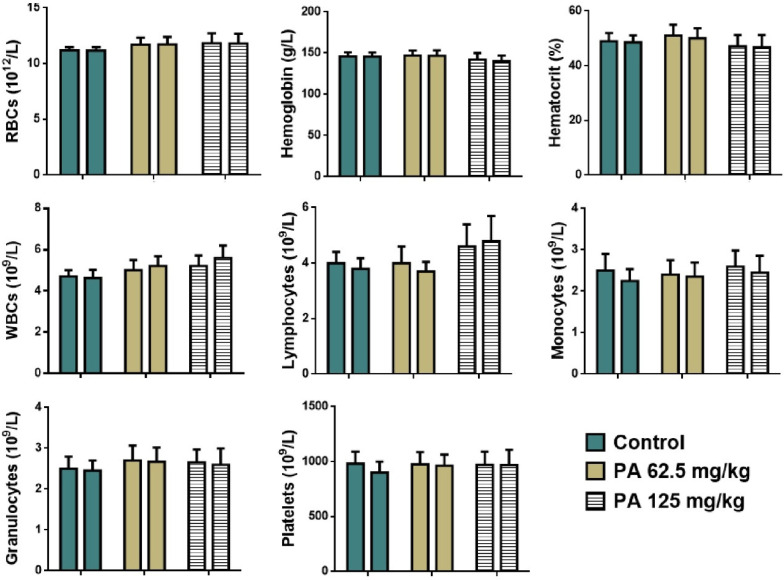
Hematological parameters of mice during the subacute toxicity study. Note: RBCs—red blood cells, WBCs—white blood cells; left column—male and right column—female.

**Figure 6 pharmaceutics-17-01040-f006:**
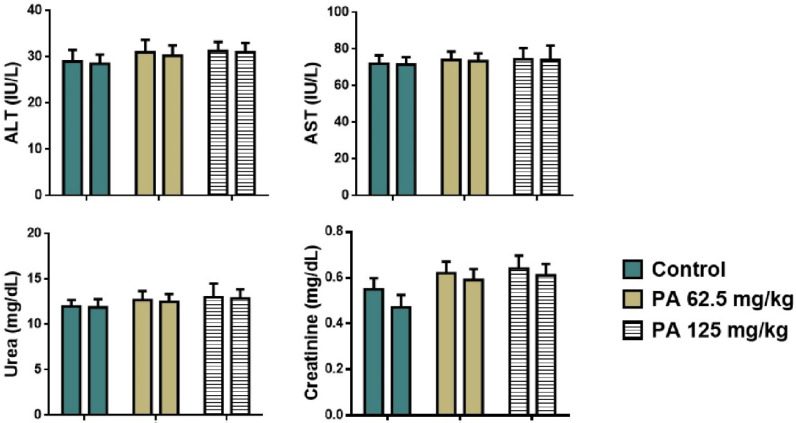
Plasma biochemistry of mice during the subacute toxicity study. Note: ALT—alaninaminotransferase, AST—aspartataminotransferase; left column—male and right column—female.

**Figure 7 pharmaceutics-17-01040-f007:**
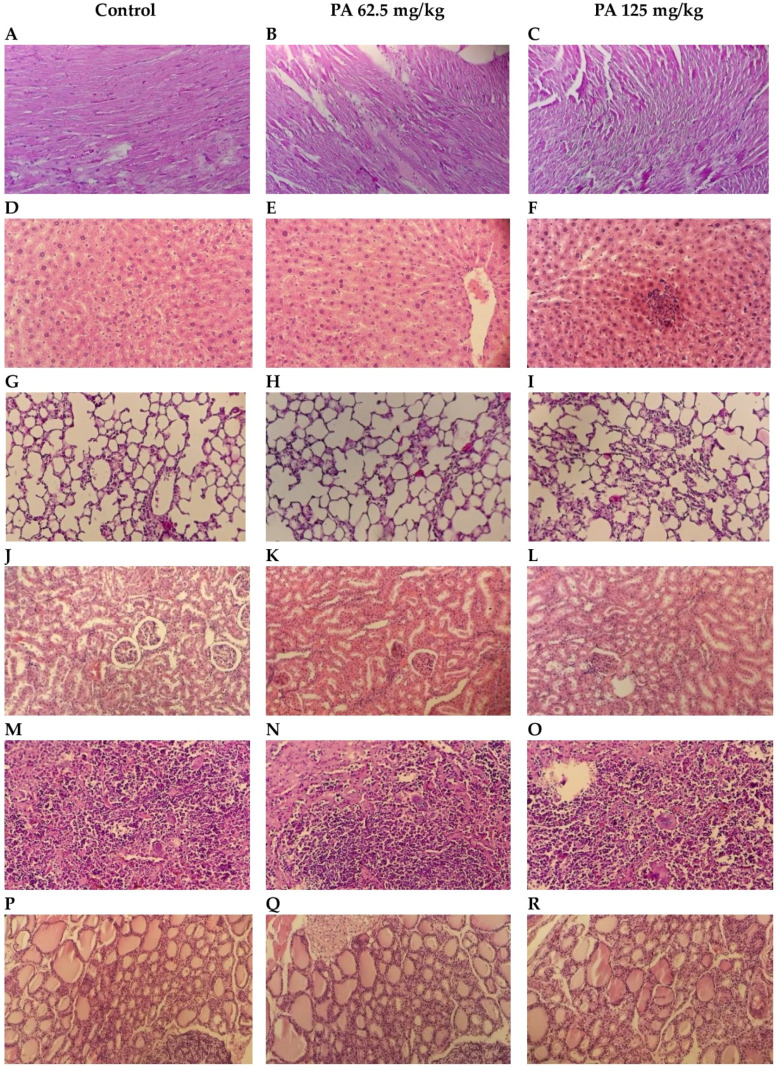
Histological structure of internal organs: heart (**A**–**C**); liver (**D**–**F**); lungs (**G**–**I**); kidneys (**J**–**L**); spleen (**M**–**O**); thyroid gland (**P**–**R**). Hematoxylin and Eosin stain; Bar, 100 µm (200× magnification).

**Figure 8 pharmaceutics-17-01040-f008:**
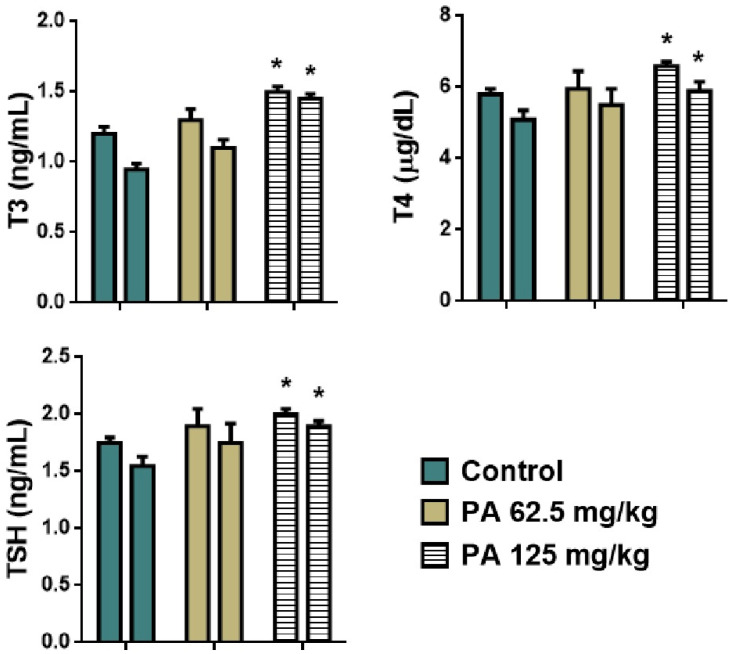
Hormones of thyroid gland of mice during the subacute toxicity study. * *p* < 0.001 compared to control. Note: T3—triiodothyronine, T4—thyroxine, TSH—thyroid stimulating hormone; left column—male and right column—female.

**Figure 9 pharmaceutics-17-01040-f009:**
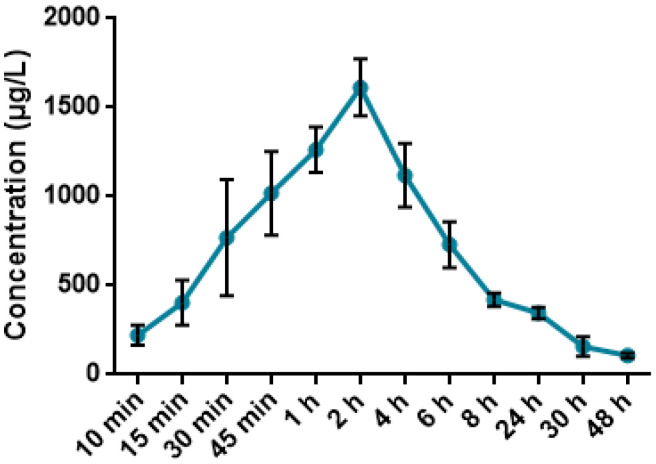
Mean plasma concentration-time profile in mice after administration of 62.5 mg/kg PA.

**Figure 10 pharmaceutics-17-01040-f010:**
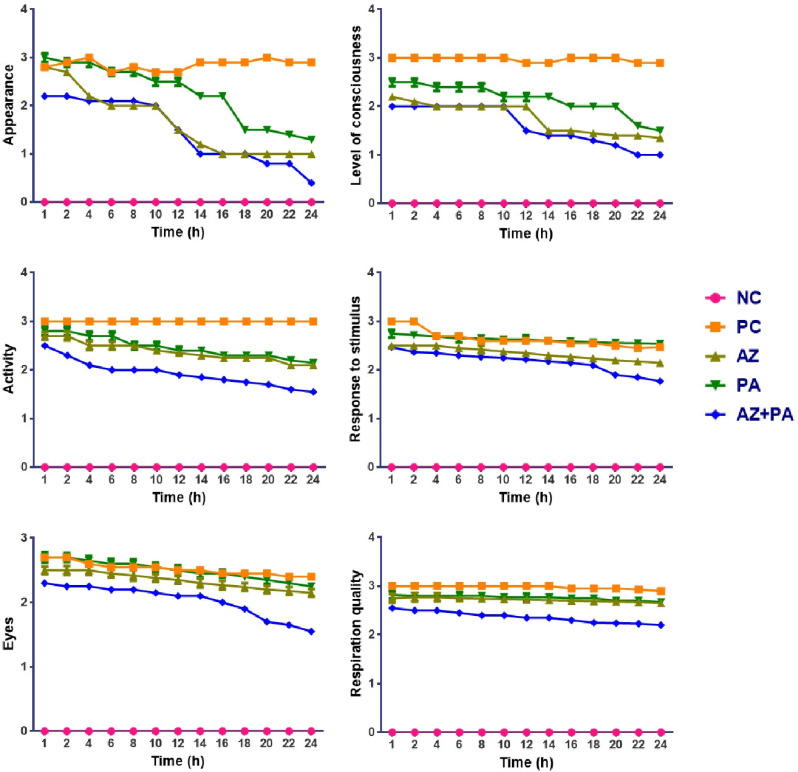
Assessment of the severity of sepsis over 24 h observation period. Note: NC—negative control, PC—positive control, AZ—group received azithromycin, PA—group received PA, AZ+PA—group received azithromycin and PA.

**Figure 11 pharmaceutics-17-01040-f011:**
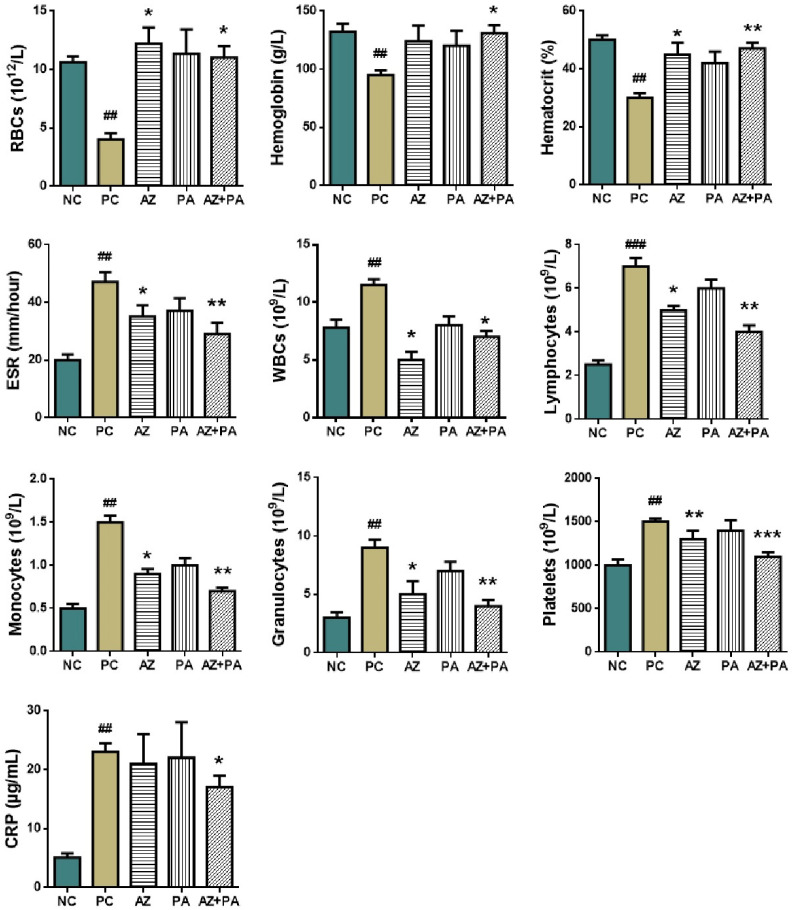
Hematological parameters of blood in mice. # *p* < 0.05, ## *p* < 0.01, ### *p* < 0.001 compared to NC and * *p* < 0.05, ** *p* < 0.01, *** *p* < 0.001 compared to PC. Note: NC—negative control, PC—positive control, AZ—group received azithromycin, PA—group received PA, AZ+PA—group received azithromycin and PA; RBCs—red blood cells, ESR—erythrocyte sedimentation rate, WBCs—white blood cells, CRP—C-reactive protein.

**Figure 12 pharmaceutics-17-01040-f012:**
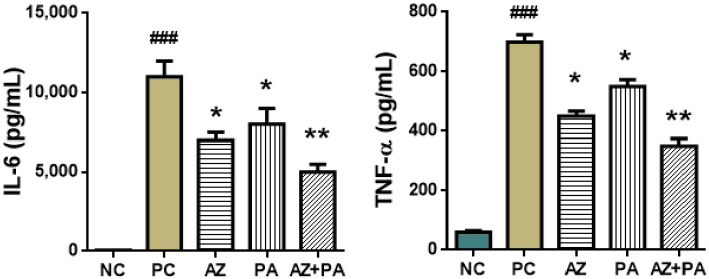
Immunological parameters of peritoneal lavage fluid in mice. ### *p* < 0.001, compared to NC and * *p* < 0.05, ** *p* < 0.01 compared to PC. Note: NC—negative control, PC—positive control, AZ—group received azithromycin, PA—group received PA, AZ+PA—group received azithromycin and PA; RBCs—red blood cells, ESR—erythrocyte sedimentation rate, WBCs—white blood cells, CRP—C-reactive protein.

**Figure 13 pharmaceutics-17-01040-f013:**
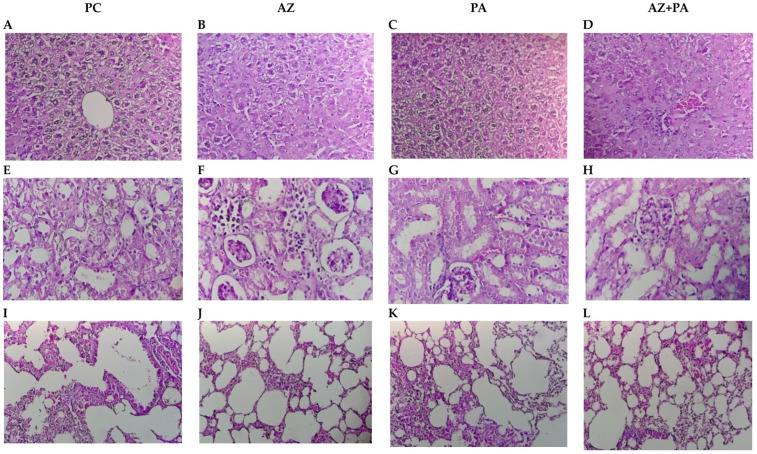
Histological structure of internal organs: liver (**A**–**D**), kidneys (**E**–**H**), lungs (**I**–**L**). Note: PC—positive control, AZ—group received azithromycin, PA—group received PA, AZ+PA—group received azithromycin and PA. Hematoxylin and Eosin stain; Bar, 100 µm (200× magnification).

**Table 1 pharmaceutics-17-01040-t001:** Physicochemical properties of PA.

Chemical Characteristics	Result
Empirical formula	C_124_H_248_O_122_NI_4_K_2_
Molecular Weight, g/mol	4288
pH	4.10
Solubility in water, g/100 mL	5 g (at 25 °C)
Kinematic viscosity, mm^2^/s	1.172
Melting temperature (°C)	168–172
Color	Dark gray
Iodine content, (g/kg)	54.65
Iodide content	82.45
Yield (%)	92

**Table 2 pharmaceutics-17-01040-t002:** Results of the stability study of PA at a temperature of 25 ± 2 °C and a humidity of 60 ± 5%.

Index Quality	Requirement R D	Storage Interval, Months				
		0	3	6	9	12
Description	Dark brown	Dark brown	Dark brown	Dark brown	Dark brown	Dark brown
Solubility in water, g/100 mL (at 25 °C)	4.5–5.0 g	4.8 g	4.7 g	5.0 g	5.0 g	5.0 g
pH	3.5–5.5	4.34	4.50	4.36	4.29	4.33
Melting point, °C	140–160	150–159	150–159	150–159	150–159	150–159
Iodine content, (g/kg)	48.96–59.84	50.65	50.63	50.49	50.43	50.45
Iodide content	71.64–87.56	79.45	77.93	79.38	78.81	79.40

**Table 3 pharmaceutics-17-01040-t003:** Body weight of mice during the acute toxicity study (g).

Day	Sex	Control	PA 2000 mg/kg	PA 5000 mg/kg
0	♂	29.72 ± 0.19	29.66 ± 0.21	29.68 ± 0.20
	♀	28.68 ± 0.43	28.76 ± 0.23	28.55 ± 0.31
3	♂	29.80 ± 0.16	29.72 ± 0.16	29.73 ± 0.19
	♀	28.76 ± 0.41	28.84 ± 0.25	28.74 ± 0.29
7	♂	29.86 ± 0.15	29.84 ± 0.17	29.81 ± 0.21
	♀	28.82 ± 0.44	28.96 ± 0.27	28.85 ± 0.25
10	♂	29.97 ± 0.12	29.90 ± 0.16	29.87 ± 0.16
	♀	28.90 ± 0.37	29.08 ± 0.30	28.98 ± 0.28

Data are presented as Mean ± SD, *n* = 10.

**Table 4 pharmaceutics-17-01040-t004:** Body weight of mice during the subacute toxicity study (g).

Day	Sex	Control	PA 62.5 mg/kg	PA 125 mg/kg
0	♂	25.7 ± 0.16	25.64 ± 0.22	25.68 ± 0.15
	♀	24.36 ± 0.33	24.18 ± 0.08	24.38 ± 0.30
7	♂	25.78 ± 0.16	25.74 ± 0.18	25.78 ± 0.08
	♀	24.44 ± 0.30	24.28 ± 0.08	24.52 ± 0.22
14	♂	25.9 ± 0.14	25.88 ± 0.16	25.92 ± 0.11
	♀	24.54 ± 0.24	24.40 ± 0.12	24.62 ± 0.16
21	♂	26.06 ± 0.13	26.08 ± 0.16	26.14 ± 0.11
	♀	24.74 ± 0.15	24.56 ± 0.11	24.8 ± 0.19
28	♂	26.22 ± 0.08	26.30 ± 0.14	26.30 ± 0.10
	♀	24.74 ± 0.15	24.74 ± 0.11	24.88 ± 0.26

Data are presented as Mean ± SD, *n* = 10.

**Table 5 pharmaceutics-17-01040-t005:** Basic pharmacokinetic parameters of mice after administration of 62.5 mg/kg PA.

Parameter	Result
AUC_total_	20,943.7 μg/mL
C_max_	2437.8 μg/mL
t_max_	4 h
t_1/2_	21.1 h
Cl_s_	0.26 L/kg
V_β_	7.8 L/kg
k_e_	0.03 h^−1^
F_abs_	91.96%
F_rel_	37.5%

**Table 6 pharmaceutics-17-01040-t006:** Body weight of mice during the experiment (g).

Day	NC (*n* = 10)	PC (*n* = 20)	AZ (*n* = 10)	PA (*n* = 10)	AZ+PA (*n* = 10)
0	22.7 ± 1.1	21.8 ± 0.7	21.5 ± 0.9	21.3 ± 0.9	22.1 ± 0.9
3	22.9 ± 1.0	17.3 ± 0.7 ^#^	18.4 ± 0.8 *	18.5 ± 1.0 *	22.0 ± 0.7 *
7	24.1 ± 1.0	16.0 ± 0.2 ^#^	17.2 ± 0.8 *	18.3 ± 1.1 *	22.2 ± 1.0 *
10	25.1 ± 1.0	15.1 ± 0.2 ^#^	17.4 ± 0.8 *	18.5 ± 0.9 *	22.5 ± 1.1 *

Data are presented as Mean ± SD. ^#^ *p* < 0.05, compared to the negative control and * *p* ≤ 0.05, compared to the positive control. Note: NC—negative control, PC—positive control, AZ—group received azithromycin, PA—group received PA, AZ+PA—group received azithromycin and PA.

## Data Availability

The original contributions presented in the study are included in the article and further inquiries can be directed to the corresponding author/s.

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
