# Peer review of "A Novel Iodine–Dextrin Complex Exhibits No Acute or Subacute Toxicity and Enhances Azithromycin Efficacy in an LPS-Induced Sepsis Model"

_pharmaceutics, 2025, doi:10.3390/pharmaceutics17081040_

Round 1
Reviewer 1 Report
Comments and Suggestions for Authors
the manuscript entitled "A Novel Iodine Complex with no Acute and Subacute Toxicity Promotes Antibiotic Potentiation Effect on LPS-induced Sepsis" has presented a new iodine complex for antibiotic potention effect. Several concerns have been addressed carefully for reconsidertion of this manuscript. For novel materials all physicochemical tests have been carefully addressed, including which methods and equipments were used for those. The most methods were missed as well as the applied supplies. This study didn't consider an in vitro study before in vivo. The investigation of in vitro assessment for instance cell line is mendotory before launch in vivo study.
Author Response
Dear Reviewer, please see the atachment

Reviewer 2 Report
Comments and Suggestions for Authors
The manuscript presents the preclinical evaluation of a novel iodine-dextrin-based semi-organic complex (PA) with potential antibiotic potentiating activity in an LPS-induced sepsis mouse model. The study encompasses acute and subacute toxicity assays, pharmacokinetic profiling, and efficacy evaluation using hematological, biochemical, cytokine, and histopathological analyses. Overall, the manuscript addresses a relevant and timely topic, especially in the context of antibiotic resistance. However, there are several issues need to be addressed before publication.
Major Comments:
- The authors should better contextualize the novelty of PA. How does this formulation compare to existing iodine-containing compounds like povidone-iodine or other dextrin-iodine complexes? A comparative discussion in the Introduction or Discussion would strengthen the manuscript.
- Mild histological changes (e.g., Kupffer cell activation, thyrocyte enlargement, stromal edema) were observed even at lower doses in subacute studies. Although not dose-limiting, these findings must be discussed in more detail rather than being downplayed.
- The manuscript requires substantial language editing. A professional language editing service is recommended.
Minor Comments:
- The title is grammatically awkward. Suggested revision:
"A Novel Iodine-Dextrin Complex Exhibits No Acute or Subacute Toxicity and Enhances Azithromycin Efficacy in an LPS-Induced Sepsis Model." - The term "semi-organic" is unusual and vague. Consider rephrasing to better reflect the chemistry.
- The discussion should better address limitations, including lack of in vitro synergy assays, absence of other antibiotic comparisons, and possible iodine-related endocrine effects.
Reviewer 3 Report
Comments and Suggestions for Authors
In this article, authors developed iodine-dextrin-based semi-organic complex (PA) and evaluated its toxicity and LPS-induced by sepsis. However, this article need major revisions to address the fo
- In introduction, add previous work done on complexes of iodine.
- Why authors choose dextran for complex formation?
- How authors confirmed the formation of complex? Please provide suitable data, which confirms the formation of complex.
- Why authors started with 2000 mg/kg for acute toxicity testing? Provide data for low or higher doses if available? What was the rational to start with 2000 mg/kg dose?
- Mention in material method section, in which form complex is administered during toxicity testing? Solution, suspension etc.?
- Why histology of liver, kidney and thyroid glands are taken? Provide images of other vital organs like lungs, heart, brain etc.
- Why only one dose i.e. 62.5 mg/kg PA is used for pharmacokinetic evaluation? Why not the other doses for comparison?
- Which dose of complex was used where authors evaluated the effect complex on sepsis? How authors rationalize dose of combination of azithromycin and complex?
- Discussion is very weak or nearly exist. Please improve the discussion with cross references.
Reviewer 4 Report
Comments and Suggestions for Authors
This study investigates the pharmacological and toxicological profile of a novel iodine–dextrin-based semi-organic complex (PA), developed as an antibiotic potentiator. The compound was evaluated through acute and subacute toxicity studies, pharmacokinetic profiling, and its therapeutic efficacy in an LPS-induced sepsis mouse model. Several important scientific and methodological questions must be addressed to fully support the compound’s potential as a safe and effective therapeutic adjuvant.
What is the stability profile of the PA complex? Given the volatile and reactive nature of iodine, data on formulation stability under physiological and storage conditions are essential.
Repeated administration of 125 mg/kg PA resulted in the enlargement of thyrocytes. Were thyroid hormone levels (e.g., T3, T4, TSH) assessed to determine endocrine implications?
In clinical settings, sepsis is typically treated after its onset. The authors administered PA prophylactically (1 hour before LPS). How do the authors justify this approach, and what are the implications for translational relevance?
Were potential pharmacokinetic interactions between PA and azithromycin investigated? Could PA influence the absorption, metabolism, or clearance of the antibiotic?
Only IL-6 and TNF-α were measured in the inflammatory response analysis. Given the complexity of the cytokine storm in sepsis, why were key cytokines such as IL-1β, IFN-γ, and IL-10 not assessed?
Did the study evaluate any systemic inflammatory markers (e.g., CRP, ESR) that would further support the observed local cytokine modulation?
How was the severity of sepsis quantified in the animal model? Was any clinical scoring system or survival analysis employed?
What is the proposed mechanism by which PA potentiates the effect of azithromycin? Is this based on modulation of immune response, enhanced antibiotic uptake, or interference with bacterial resistance mechanisms?
Has the potential of PA to potentiate other antibiotics (beyond azithromycin) been explored or hypothesized?
Round 2
Reviewer 1 Report
Comments and Suggestions for Authors
The concerns have been addressed carefully.
Reviewer 4 Report
Comments and Suggestions for Authors
The authors have made substantial changes in several parts of the paper and addressed the reviewers’ comments. This manuscript may be accepted for publication.